# Synergistic effects of *Rhynchosia nulubilis* and *Polygonum multiflorum* extract combination on cell proliferation via targeting IGFBP-1 & NT-3 and cytotoxicity suppression in testosterone-induced human dermal papilla cells

Jiwon Seo[1], Chanhyeok Jeong[1], Jung Han Yoon Park [2,3], Chang Hyung Lee[2,4,5]*, Ki Won Lee [1,2,3,6,7]*

**1** Department of Agricultural Biotechnology and Research Institute of Agriculture and Life Sciences, Seoul National University, Seoul, Republic of Korea, **2** Bio-MAX Institute, Seoul National University, Seoul, Republic of Korea, **3** Advanced Institutes of Convergence Technology, Seoul National University, Suwon, Republic of Korea, **4** School of Pharmacy, Sungkyunkwan University, Suwon, Republic of Korea, **5** John A. Paulson School of Engineering and Applied Sciences, Harvard University, Allston, Massachusetts, United States of America, **6** Institutes of Green Bio Science & Technology, Seoul National University, Pyeongchang, Republic of Korea, **7** Department of Agricultural Biotechnology and Center for Food and Bioconvergence, Seoul National University, Seoul, Republic of Korea

☯ These authors contributed equally to this work.
* changhyung@skku.edu (CHL); kiwon@snu.ac.kr (KWL)

## Abstract

This study examines the synergistic effects of extracts from *Rhynchosia nulubilis* (RN) and *Polygonum multiflorum* (PM) on the proliferation of human dermal papilla cells (hDPCs) and the alleviation of testosterone-induced cytotoxicity. Human dermal papilla cells (hDPCs) were treated with varying concentrations of RN and PM extracts, administered both individually and in multiple combinations at different ratios. The findings indicated that a 4:1 combination of RN and PM extracts significantly enhanced hDPC proliferation relative to the individual extracts, particularly in the presence of testosterone, which induced cytotoxicity. A significant synergistic effect was observed at a 4:1 ratio, resulting in the creation of a human growth factor array to identify targets associated with this synergy. The combined-extract group exhibited elevated levels of two significant growth factors: insulin-like growth factor-binding protein-1 (IGFBP-1) and neurotrophin-3 (NT-3). This was additionally validated through Western blot analysis. HPLC analysis identified six compounds and screening was conducted. As a result, genistein derived from RN and 2,3,5,4′-tetrahydroxystilbene-2-O-β-D-glucoside (2354-T2G) sourced from PM may be responsible for these effects. This is the first study to illustrate the significant synergistic effect of the combination of RN and PM, suggesting a potential treatment strategy that boosts the efficacy of natural compounds through synergy. The results suggest that the combined extracts could be useful as an effective treatment strategy

**Data availability statement:** All underlying raw data including values behind means and standard deviations, data points used to build graphs, and points extracted from images are provided in the Supporting Information files and deposited in a stable public repository (URL: https://figshare.com/s/c3eada5cbad9525169a4).

**Funding:** This work was supported by BOBSNU Co. Ltd. (to CHL); the Bio & Medical Technology Development Program of the National Research Foundation (NRF), funded by the Korean government (MSIT) (No. 2020M3H1A1073304 to CHL); the National Research Foundation of Korea (NRF) grant funded by the Korea government (MSIT) (RS-2024-00333238 to CHL); Korea Institute of Planning and Evaluation for Technology in Food, Agriculture and Forestry (IPET) through Agriculture and Food Convergence Technologies Program for Research Manpower Development, funded by the Ministry of Agriculture, Food and Rural Affairs (MAFRA) (RS-2024-00402136 to KWL); and Korea Institute of Planning and Evaluation for Technology in Food, Agriculture and Forestry (IPET) through Technology Commercialization, funded by the Ministry of Agriculture, Food and Rural Affairs (MAFRA) (IP122036 to KWL). The funders had no role in study design, data collection and analysis, decision to publish, or preparation of the manuscript.

**Competing interests:** The authors have declared that no competing interests exist.

for hair loss and associated disorders. Furthermore, this synergy-based approach has potential applications in future research on natural products.

## Introduction

Hair loss, or alopecia, is a prevalent issue affecting millions of individuals worldwide [1]. Hair loss imposes economic burdens due to treatment costs and loss of productivity, and also leads to emotional distress, such as decreased self-esteem, social anxiety, and even depression. These psychological effects can significantly diminish the quality of life for affected individuals and have broader societal implications. The condition may arise from multiple factors, including genetic predisposition, environmental influences (e.g., pollution and ultraviolet radiation), hormonal changes related to aging or endocrine disorders, nutritional deficiencies, specific medications, and stress [2]. Different types of alopecia, such as androgenetic alopecia, alopecia areata, and telogen effluvium, present unique challenges in diagnosis and treatment, making the search for effective therapies more critical. Various therapeutic methods and approaches for these conditions have recently been developed and investigated, with a notable increase in research focused on dermal papilla cells, which play a crucial role in hair growth and hair loss.

Human dermal papilla cells (hDPCs) play an important role in hair development and growth [3]. They are the integral components of hair follicles, essential for regulating the hair growth cycle and forming new hair shafts [4]. Located at the base of hair follicles within the dermal papilla, hDPCs interact closely with surrounding epithelial cells and induce hair follicle morphogenesis during embryonic development. They secrete various signaling molecules, such as growth factors that influence hair follicle formation and the cycling between anagen, catagen, and telogen phases. Due to their role in hair biology, hDPCs have gained significant attention in research related to hair loss. Hair loss involves hair shedding and thinning, often associated with hDPC abnormal damage, premature senescence, or dysfunction [5]. Alterations in the activity or viability of these cells can disrupt the hair growth cycle, leading to conditions such as follicular miniaturization and reduced hair density commonly observed in pattern baldness.

In human hair follicles, testosterone exerts its influence directly by binding to androgen receptors and affecting gene expression related to hair growth [6]. Previous studies have demonstrated elevated testosterone levels in individuals with hair loss, particularly in cases of male-pattern baldness [7]. This suggests that the scalp affected by baldness has an increased testosterone sensitivity that can negatively impact hair follicle function [8]. Testosterone induces changes in hDPCs that lead to hair follicle miniaturization and shortens the anagen phase, significantly reducing the anagen-to-telogen ratio [9]. Furthermore, treatment with testosterone affects hDPC proliferation and apoptosis. Testosterone can induce cell death and inhibit cell proliferation, contributing to the weakening and eventual loss of hair follicles. This effect underscores the critical role of androgens, particularly testosterone, in hair follicle health and the progression of hair loss conditions, such as androgenetic alopecia.

Therefore, investigating the effects of testosterone on hDPC proliferation and apoptosis is crucial for understanding the underlying mechanisms of hair loss.

Upon examination of current pharmaceutical products and functional cosmetics (alternatives), it becomes evident that many of these products exhibit low efficacy or can induce adverse side effects in users. Merck's Propecia, a widely pre-scribed medication for hair loss treatment approved by the U.S. Food and Drug Administration, functions by reducing dihy-drotestosterone (DHT) levels, a derivative of male hormones that can trigger hair loss, and inhibiting hair loss progression [10]. However, it is associated with common side effects, such as erectile dysfunction and reduced libido, and is unsuit-able for premenopausal women [11]. The prolonged use of minoxidil may lead to side effects, such as facial redness, chest palpitations, and arrhythmias [12]. Furthermore, some currently available functional cosmetics aimed at improving hair loss fail to provide sufficient efficacy or offer appropriate solutions. To address these issues, a safe and effective alter-native is required, and natural ingredients might be one excellent option for preventing and alleviating hair loss symptoms. In this situation, synergy has been suggested for a long time to get the most out of safe, naturally derived materials, and research in this area has been ongoing and published.

Synergy refers to the phenomenon where the combined effects of two or more bioactive compounds (or extracts) are greater than the sum of their individual effects [13]. Synergistic effects of natural ingredient combinations have been stud-ied in various fields, including cancer, cardiovascular diseases, and diabetes [14–16]. First, we conducted a series of tests on a variety of natural extracts to identify those that were generally safe and effective as the concentration increased, as indicated by the findings of previous studies. Two promising candidates, *Rhynchosia nulubilis* (RN) and *Polygonum multi-florum* (PM), have emerged as the most effective and safest options for further investigation through this process. RN has demonstrated its ability to promote hDPC proliferation and expand blood vessels, facilitating hair growth [17]. PM extracts support hair growth by extending the anagen phase and counteracting the effects of androgens [18]. Specific compounds identified in the PM extract, such as 2,3,5,4′-tetrahydroxystilbene-2-O-$\beta$-D-glucoside (2354-T2G) and emodin, have been reported to promote hair growth [19]. Although various studies have investigated the physiological activities of RN and PM, research on the synergistic effects of their combination is still unknown. Therefore, this study aimed to elucidate the synergistic effects of the RN and PM extract combinations on hDPC proliferation and growth factor secretion and deter-mine the optimal combination ratio.

## Materials & methods

### Chemicals and extract preparation

Dulbecco's modified Eagle's medium (DMEM) and fetal bovine serum (FBS) were purchased from Gibco (Carlsbad, CA, USA). Penicillin (10,000 IU) and streptomycin (10,000 µg/mL) were purchased from Mediatech (Manassas, VA, USA). 3-(4,5-Dimethylthiazol-2-yl)-2,5-diphenyltetrazolium bromide (MTT) was purchased from Affymetrix/USB (Cleveland, OH, USA). Dimethyl sulfoxide (DMSO), testosterone (dissolved in DMSO), minoxidil (dissolved in 0.12 mM HCl), daidzein, glycitein, and genistein were purchased from Sigma-Aldrich (St. Louis, MO, USA). 2354-T2G was purchased from TCI (Tokyo, Japan). Emodin-8-O-$\beta$-D-glucoside (E8G) and physcion 1-O-$\beta$-D-glucoside (P8G) were purchased from Chem-Faces Co., Ltd. (Wuhan, China). BOBSNU Co., Ltd. (Suwon, Republic of Korea) manufactured and provided the RN and PM extracts. Extraction was conducted using a 70% ethanol solution at 31.5°C for 15 hours. The solvent was evaporated, and the product was freeze-dried at −20°C.

### Cell proliferation and cytotoxicity assay

The cells (hDPCs) were purchased from CEFO Bio Co., Ltd. (Seoul, Republic of Korea) and incubated at 37°C in a 5% $CO_2$ humidified atmosphere. To investigate the effects of the RN and PM extracts on hDPC proliferation, an MTT cell proliferation assay was conducted. hDPCs were seeded at a density of $3 \times 10^4$ per well in a 96-well plate and cultured for 24 hours. hDPCs were treated with various concentrations (0–80 µg/mL) of the RN and PM extracts and incubated for 48

hours. The MTT solution was added to the medium, and DMSO was used to fully dissolve the formazan crystals. Absorbance was measured at 540 nm using an Epoch microplate spectrophotometer (Winooski, VT, USA). Cells in DMEM with 10% FBS were seeded into a 96-well plate at a density of $3 \times 10^4$ per well. Cells were cultured for 24 h, and the medium was changed to serum-free DMEM for starvation. After an additional incubation at 37°C for 24 h, hDPCs were treated with samples in serum-free medium for 1 hour, after which 200 μM testosterone was added and incubated for 72 hours.

### Human growth factor array

Screening for their effects on growth factor expression, the Abcam Human Growth Factor Antibody Array kit (ab134002) was used to simultaneously detect 41 human growth factors. hDPCs ($3 \times 10^4$ per dish) were seeded in a 10 cm dish and cultured overnight. The medium was changed to serum-free DMEM. Cells were treated with RN, PM, and the combined extracts (at a 4:1 ratio) for 48 h, and proteins were collected for growth factor analysis. The array was processed following the manufacturer's instructions. Protein expression was detected by the AE-9150 Ez-Capture II device (Atto, Tokyo, Japan).

### Western blotting

To detect protein expression, cells were lysed with a cell lysis solution consisting of 10 mM Tris (pH 7.5), 150 mM NaCl, 5 mM EDTA, 0.1% Triton X-100, 1 mM dithiothreitol, 0.1 mM phenylmethylsulfonyl fluoride, and 10% glycerol. Cell lysates were then harvested and centrifuged at 13,572×g for 10 minutes. Protein concentrations in the lysates were quantified using the BCA protein assay kits (Thermo Scientific, Waltham, MA, USA) according to the manufacturer's instructions. For protein separation, the samples underwent electrophoresis in a 10% sodium dodecyl sulfate-polyacrylamide gel electrophoresis system and were transferred to polyvinylidene difluoride membranes (Millipore, Bedford, MA, USA). Membrane blocking was performed using 3% skim milk for 1 hour, followed by overnight incubation at 4°C with the designated primary antibody. After incubation with horseradish peroxidase-conjugated secondary antibody(Life Technologies, Waltham, MA, USA), protein bands were detected by the AE-9150 Ez-Capture II device (Atto, Tokyo, Japan).

### Quantification of active compounds

The samples and compounds were examined using a Waters high-performance liquid chromatography (HPLC) system (Milford, MA, USA). This system was equipped with a Waters 2998 photodiode array detector and employed an HPLC Phenomenex Luna C18 column (5 μm, 100 Å, dimensions $250 \times 100$ mm²; Phenomenex, Torrance, CA, USA). The column temperature was held at 24°C, and chromatograms were processed at a wavelength of 254 nm. A gradient solvent system separated the components: solvent A (0.1%, v/v aqueous formic acid) and solvent B (0.1%, v/v formic acid in acetonitrile). The flow rate was maintained at 1 mL/min, following the specified procedure: 0–2 minutes, transitioning from 0% to 5% of solvent B, and 2–15 minutes, transitioning from 5% to 100% of solvent B. HPLC-grade solvents (Duksan, Daejeon, Republic of Korea) were used for HPLC analysis.

### Statistical analysis

All data are presented as mean ± standard deviation (SD). Statistical analyses were performed using SPSS version 12.0 (SPSS, Chicago, IL, USA). Comparisons between two groups were analyzed using Student's t-test.

## Results

### Effects of individual RN and PM extracts on hDPC proliferation under normal and testosterone-induced conditions

The cells were treated with varying concentrations of the RN extract (2.5, 5, 10, 20, 40, and 80 μg/mL), and cell proliferation rates were measured after 48 hours of incubation. Fig 1a shows a 61% increase in proliferation at 80 μg/mL

concentration compared to control. Similarly, PM extract treatment led to a dose-dependent increase in hDPC proliferation, with an impressive 509% increase at 80 μg/mL concentration. Under testosterone conditions, RN showed protective effects at higher concentrations (40 and 80 μg/mL), while PM demonstrated effectiveness across a broader range (10–80 μg/mL) (Fig 1b). These results indicated that the RN and PM extracts are natural ingredients with potential for hair growth promotion. Based on these findings, we investigated whether combining these extracts at lower, minimally effective concentrations could produce synergistic effects under testosterone-induced conditions.

## Synergistic effects of the combined extracts in testosterone-induced cytotoxicity

This study established a natural product screening model that reduces damage to hDPCs induced by testosterone, a precursor to DHT, which acts on the scalp [7]. Under testosterone conditions, individual treatments with RN, PM, or minoxidil showed minimal effects on hDPC proliferation. Cell survival rates compared to control were 54% for RN treatment, 59% for PM treatment, and 31% for minoxidil treatment. However, the combined extracts at a 4:1 ratio (RN 20 μg/mL: PM 5 μg/mL) demonstrated exceptional hDPC proliferation even in the presence of testosterone (Fig 2). In the presence

**(a)**

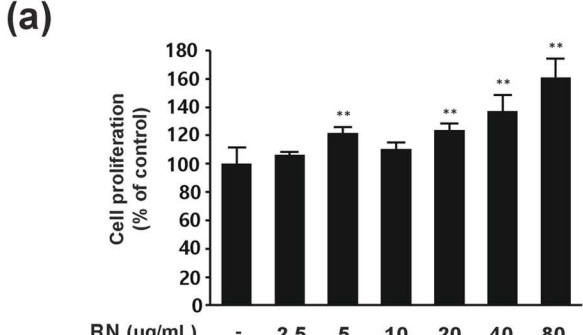 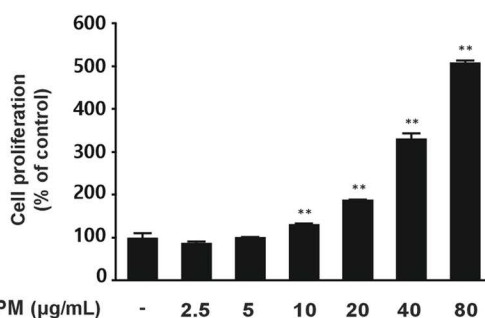

**(b)**

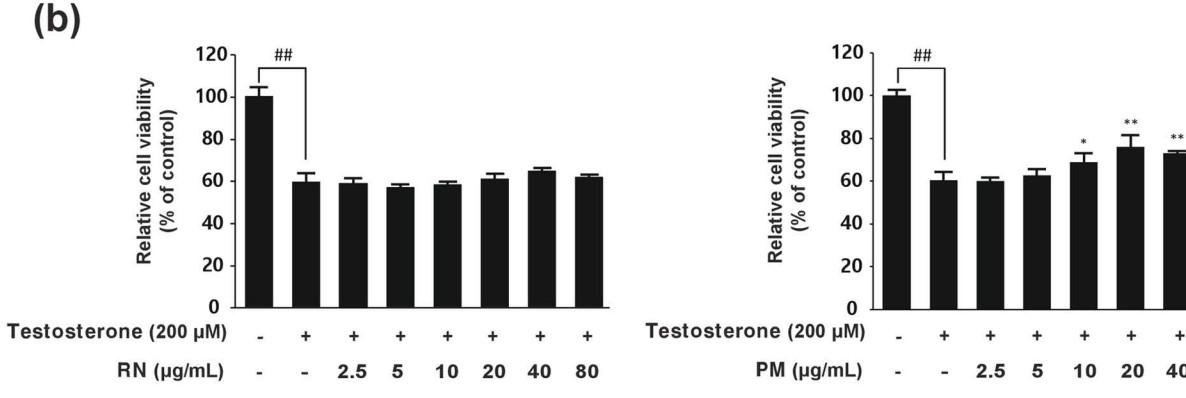

**Fig 1. Effects of RN and PM on hDPC proliferation under normal and testosterone-induced conditions.** (a) Relative viability values of hDPCs treated with various concentrations (0-80 μg/mL) of RN and PM for 48 h under normal conditions, obtained by the MTT assay. Data are presented as the mean ± SD. *$p < 0.05$; **$p < 0.01$ compared to control. (b) Cell viability of hDPCs treated with RN and PM under testosterone-induced conditions (200 μM). Data are the mean ± SD. *$p < 0.05$; **$p < 0.01$ compared to the testosterone group.

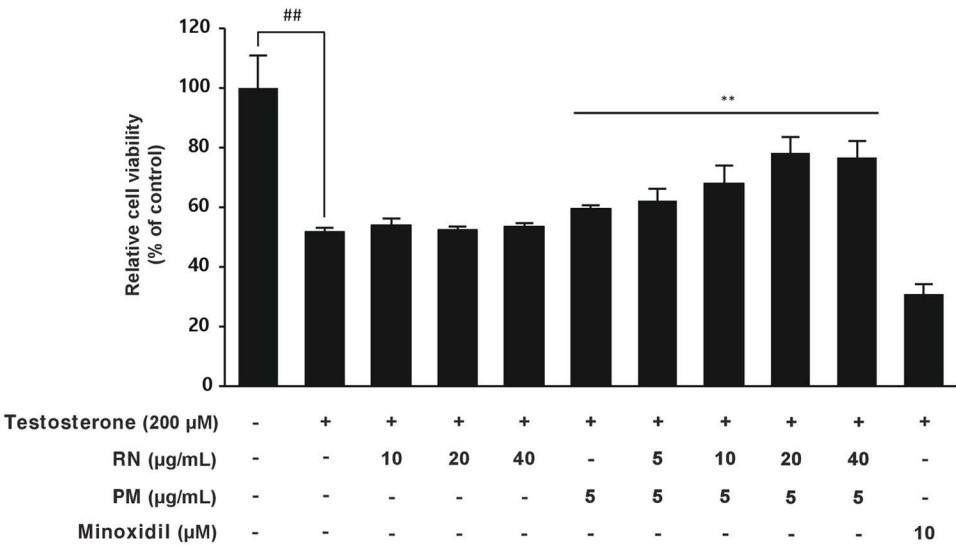

**Fig 2. Synergistic effects of the combined extracts on hDPCs under testosterone-induced conditions.** Viability of hDPCs treated with individual extracts, minoxidil, and combined extracts (4:1 ratio) under testosterone-induced conditions, obtained by the MTT assay. Data are the mean ± SD. n > 3 for each group. **$p < 0.01$ compared to the testosterone group.

of testosterone, the group treated with the combined extracts showed a substantial increase in cell viability from 52% to 78%, demonstrating that the combined extracts have superior synergistic effects compared to the individual extracts, even in the testosterone-treated condition.

## Growth factor expression analysis in hDPCs treated with individual and combined extracts

Based on the synergistic effects observed under testosterone-induced conditions, we investigated whether the 4:1 ratio combination would affect growth factor expression. Proteins collected after 4 days of culture with RN, PM, or their combination were analyzed for 41 growth factors using a human growth factor antibody array (Fig 3a). The analysis revealed a substantial increase in the secretion of insulin-like growth factor-binding protein-1 (IGFBP-1) and neurotrophin-3 (NT-3) specifically in the group treated with the combined extracts of RN and PM at the 4:1 ratio (RN 20 µg/mL: PM 5 µg/mL). Quantification of the array results showed that while individual treatments had minimal effects, the combined treatment significantly enhanced the secretion of both IGFBP-1 and NT-3 (Fig 3b). These findings suggested that the synergistic effects of the combined extracts involve the enhanced secretion of these key growth factors associated with hair growth.

## Western blot validation of IGFBP-1 and NT-3 expression

To further confirm the array results, we examined IGFBP-1 and NT-3 protein expression levels using Western blot analysis. In agreement with the array data, IGFBP-1 levels, which were barely detectable in the control group, showed a significant increase in the RN/PM (4:1) combination group (Fig 4a). Similarly, NT-3 expression was markedly elevated in the combination group compared to individual extract treatments (Fig 4b). Quantitative analysis of the Western blot results further validated that the combined treatment significantly enhanced the expression of both growth factors. These findings provided additional confirmation that the synergistic effects of the RN/PM combination are associated with increased production of IGFBP-1 and NT-3, two growth factors crucial for hair follicle development.

                                                            

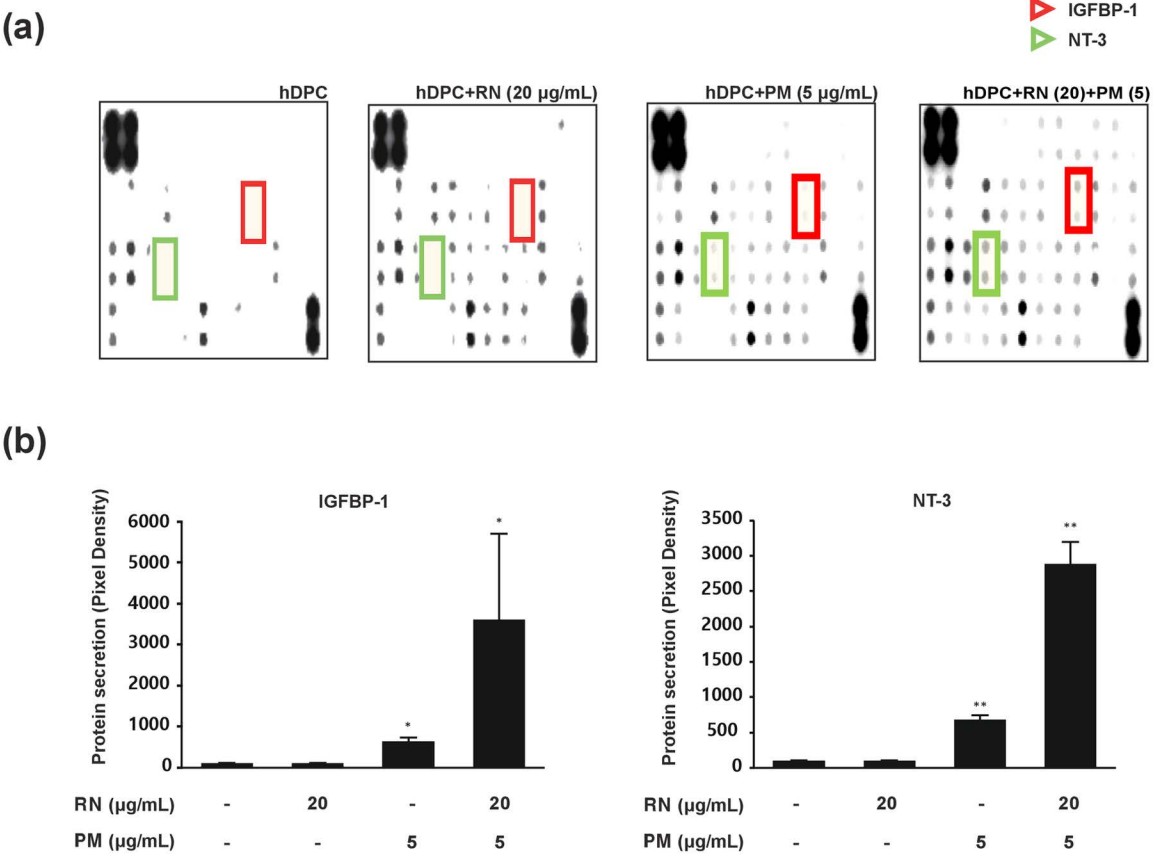

**Fig 3. Analysis of growth factor expression in hDPCs treated with RN and PM extracts. (a)** Representative images of growth factor antibody array analysis in proteins obtained from hDPCs treated with individual extracts and combined extracts (4:1 ratio). **(b)** Quantification of IGFBP-1 and NT-3 expressions from array analysis. Data are the mean±SD. *$p < 0.05$; **$p < 0.01$ compared to control.

## Quantification of the active compounds in the RN and PM extracts

The major functional components of RN are aglycone isoflavones, such as daidzein, glycitein, and genistein. (Fig 5a) [20–22]. According to a report by the Herbal Medicine Research Division of the Korea Ministry of Food and Drug Safety, the major active components of PM are 2354-T2G, E8G, and P8G (Fig 5b) [23–25]. The content analysis of these compounds in RN and PM was conducted using a quantitative method based on standard curves at 254 nm. HPLC analysis of the 70% ethanol extract of RN revealed 0.188 µg/mg at daidzein (RT: 8.16 min, MW 254.23), 1.163 µg/mg at glycitein (RT: 8.37 min, MW 284.26), and 0.541 µg/mg at genistein (RT: 9.18 min, MW 270.24). Similarly, the PM extract contained 2354-T2G (RT: 7.14 min, MW 406.38) at 142.771 µg/mg, E8G (RT: 8.02 min, MW 432.38) at 2.272 µg/mg, and P8G (RT: 8.44 min, MW 446.41) at 4.836 µg/mg (Table 1).

## Effects of the active compounds on hDPC proliferation

Based on the quantitative analysis of compounds in the RN and PM extracts, this study examined whether those compounds had efficacy in hDPC proliferation under testosterone conditions. Under testosterone conditions, it dramatically affected hDPC proliferation (cell survival rates were 40%). However, genistein and 2354-T2G demonstrated exceptional hDPC proliferation even in the presence of testosterone. When hDPCs were exposed to testosterone and treated with

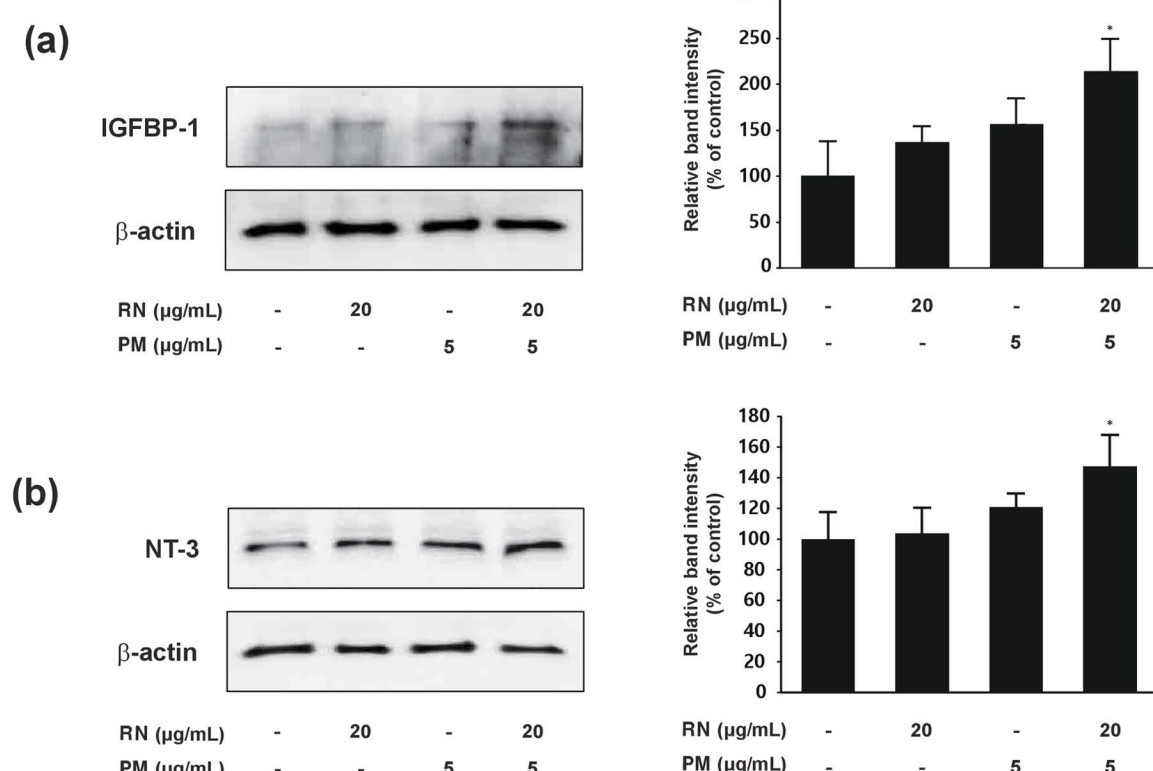

**Fig 4. Validation of IGFBP-1 and NT-3 expressions by Western blot analysis.** (a) Western blot analysis and quantification of IGFBP-1 protein expression in hDPCs treated with individual extracts and combined extracts (4:1 ratio). (b) Western blot analysis and quantification of NT-3 protein expression in hDPCs treated with individual extracts and combined extracts (4:1 ratio). Data are the mean ± SD from three independent experiments. *$p < 0.05$ compared to control.

genistein, cell viability improved from 38% to 58% (Fig 6a), and 2354-T2G significantly increased cell viability from 41% to 61% under testosterone-induced conditions (Fig 6b). Considering both the concentration of each compound in the extracts and their efficacy, genistein from RN and 2354-T2G from PM were identified as the main active ingredients responsible for promoting hDPC proliferation under testosterone-induced conditions.

## Discussion

Dermal papilla cells are well known for their critical role in regulating hair growth and are essential in hair formation and the hair growth cycle [26]. Located at the base of the hair follicle, they interact closely with surrounding epithelial cells to initiate and maintain hair follicle development. They generate numerous growth factors and cytokines, such as vascular endothelial growth factor, IGF-1, and basic fibroblast growth factor, significantly influencing the proliferation, differentiation, and function of neighboring cells [3]. This complex cellular interaction becomes particularly significant in male-pattern hair loss, the most common form of alopecia, which is primarily androgen-dependent with testosterone and DHT as prominent factors [27]. Androgen hormones, such as testosterone, inhibit hair follicle activity during the early stages of hair growth [9], and this study established a model condition similar to hair loss by treating hDPCs with testosterone [7]. Topical application and systemic administration vary in their mechanisms of action and therapeutic goals. Systemic approaches, including oral medications for hair loss, primarily target hormonal pathways associated with the condition. In contrast, topical treatments aim to improve blood circulation in the scalp and hair follicles while stimulating growth factors. This study

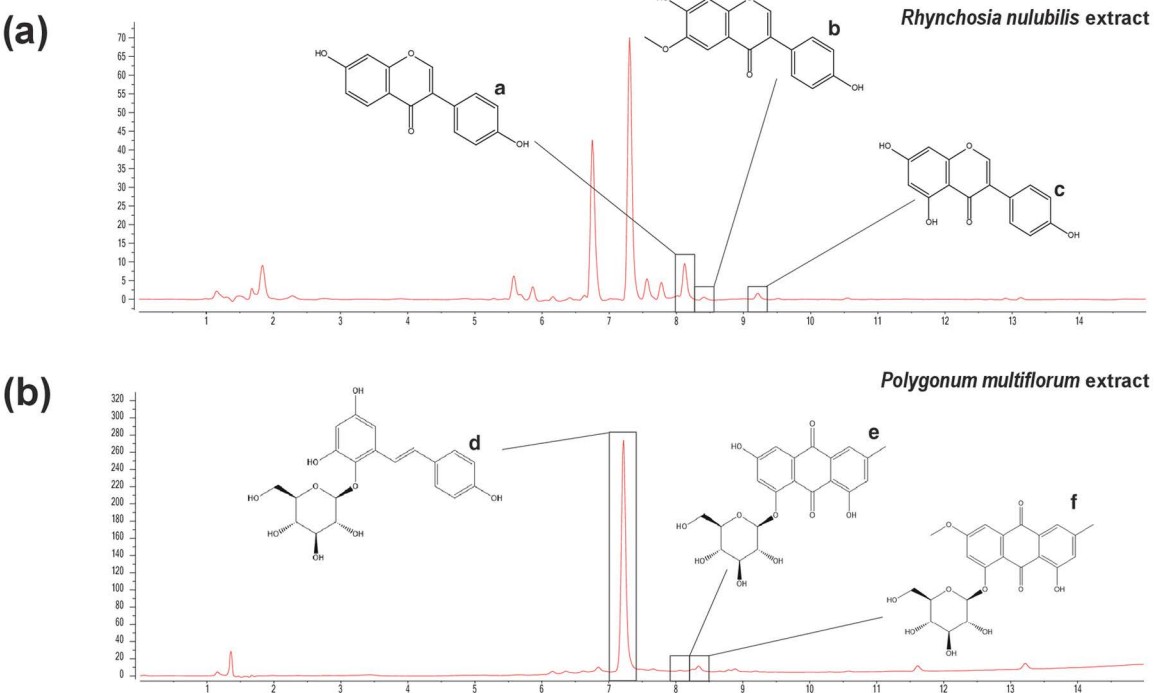

**Fig 5. Chromatogram of the active compounds in RN and PM as determined using HPLC and LC-mass spectrometry analyses.** (a) HPLC chromatogram of the RN extract at 254 nm showing peaks for daidzein (*a*), glycitein (*b*), and genistein (*c*). (b) HPLC chromatogram of the PM extract at 254 nm showing peaks for 2354-T2G (*d*), E8G (*e*), and P8G (*f*).

**Table 1. Quantification of the major compounds of ethanol extract from RN and PM by HPLC.**

| Peak | Compound | µg/mg | M.W. | RT |
|---|---|---|---|---|
| a | Daidzein | 0.188 µg/mg | 254.23 | 8.16 min |
| b | Glycitein | 1.163 µg/mg | 284.26 | 8.37 min |
| c | Genistein | 0.541 µg/mg | 270.24 | 9.18 min |
| d | 2354-T2G | 142.771 µg/mg | 406.39 | 7.14 min |
| e | E8G | 2.272 µg/mg | 432.38 | 8.02 min |
| f | P8G | 4.836 µg/mg | 446.41 | 8.44 min |

aimed to screen natural extracts or compounds that promote hair growth by activating growth factors while maintaining stable hormone levels, thus identifying potential candidates for topical application.

In this study, minoxidil was selected as a positive control based on its established role in hair growth research and its relevance to topical treatment development. While finasteride specifically targets the androgen pathway through 5α-reductase inhibition, minoxidil has been widely used in hDPC studies due to its direct effects on cell proliferation and vasodilation. [28,29]. However, in our testosterone-induced condition, minoxidil treatment led to decreased cell viability, possibly due to additional cellular stress in already testosterone-stressed hDPCs. This outcome suggests complex interactions between different treatment mechanisms in hormonal environments and warrants further investigation of cellular stress responses, which may be an important consideration in developing hair loss treatments. Since our model was initially designed to emphasize topical mechanisms over systemic ones, it is scientifically reasonable that no significant efficacy would be detected in a testosterone-induced model. This outcome was replicated in our in vitro model, suggesting

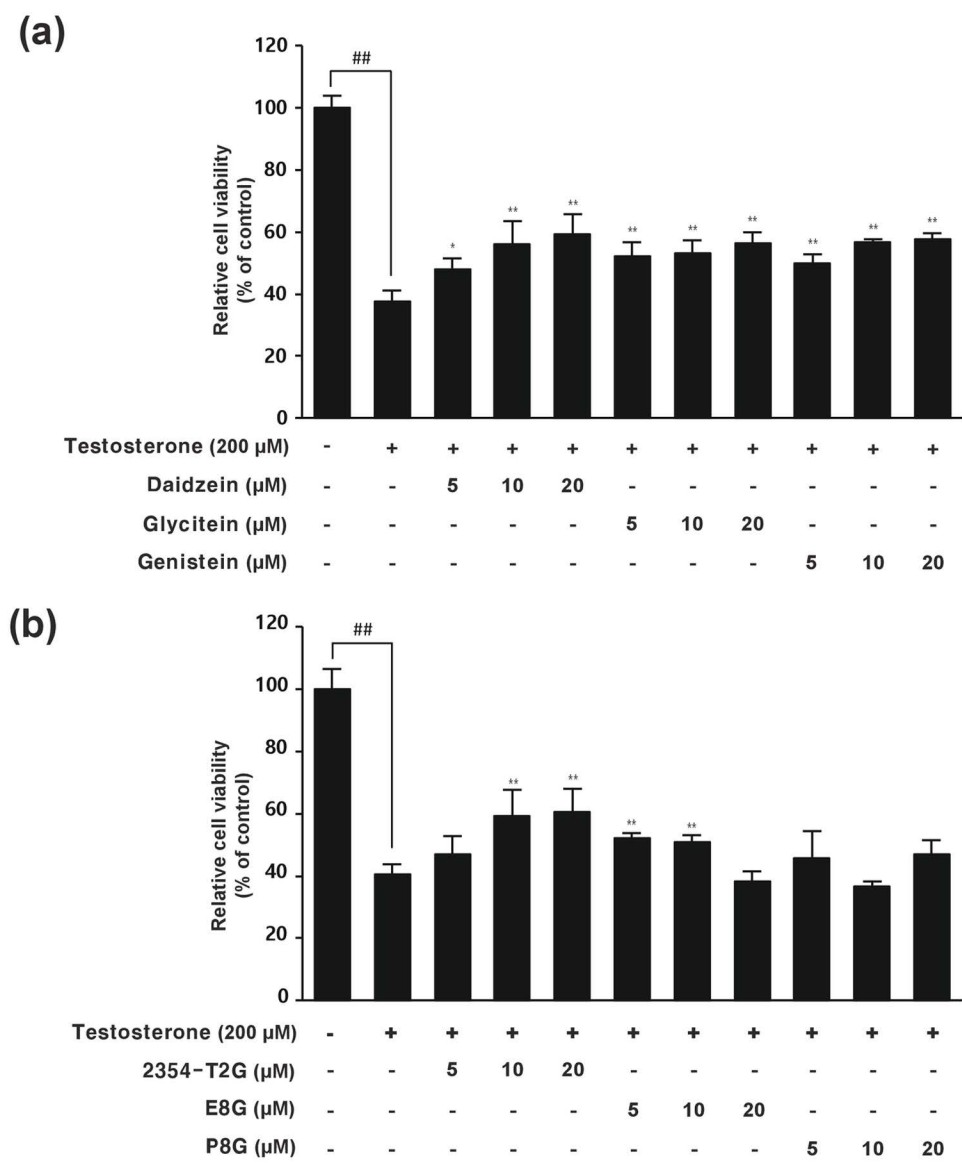

**Fig 6. hDPC viability enhancement by active compounds from RN and PM under testosterone-induced conditions.** (a) hDPC viability after treatment with daidzein, glycitein, and genistein in testosterone-induced conditions. (b) Cell viability effects of 2354-T2G, E8G, and P8G on hDPCs under testosterone-induced conditions. Cell viability was determined by the MTT assay. Data are the mean ± SD. *$p < 0.05$; **$p < 0.01$ compared to the testosterone group.

potential advantages of the natural extracts in testosterone-dominant conditions. These findings provide a foundation for further investigation into natural extract combinations as alternative approaches for treating androgen-dependent hair loss.

As we mention in the introduction, synergy is the effect that occurs when different compounds are mixed together that is stronger than the effect of either compound alone. A previous study that used the Chou-Talalay method showed the best ways to study chemical (drug) combinations and introduced the combination index (CI) to measure synergy [30]. Recent synergy research has focused on improving current medications or mitigating negative effects by reducing dosages through co-administration. In this study, we focus on natural extracts, generally regarded as having high safety profiles

due to their long history of use and documented consumption across various cultures. However, these extracts are often less effective and less specific to their targets because of the complexity of these extracts, which contain numerous phytochemicals [31,32]. This complexity presents a significant challenge compared to single-compound therapies. For this reason, effectively harnessing synergy concepts and methodologies may overcome this limitation by offering substantial efficacy with comparatively low toxicity and thus broadening clinical potential. Nevertheless, most studies of natural materials, including our previous study, have evaluated each compound's efficacy individually, often relying on a combination index for quantification [16,33]. However, upon revisiting the concept, we found that an observed synergistic effect still holds true, even if we cannot derive a CI value from EC50.

Using an hDPCs model, we evaluated plant-derived extracts for both their hair growth-promoting and testosterone-protective effects. Two candidates (RN, PM) were found to be the most effective at the highest concentration in our preliminary result. Subsequent experiments were conducted by combining non-efficacious concentration conditions, utilizing prior methodologies that recognize synergy even in situations where CI-based calculations are impractical. We examined the combination of two distinct extracts, RN and PM, by considering two main strategies: increasing the ratio of RN or increasing the ratio of PM. In the experiment, the ratio of extract RN, which showed less efficacy at lower concentrations compared to extract PM, was slowly raised from 1:1–8:1 to test the synergistic effect. Synergy was initially observed at a 2:1 ratio, with a markedly enhanced effect at a 4:1 ratio. The efficacy at a ratio of 8:1 was slightly lower than at 4:1, suggesting a possible saturation effect of synergism (Fig 2). Subsequently, a growth factor array analysis was performed to identify the specific targets that were upregulated under the 4:1 condition. Screening of 41 growth factors using the human growth factor array kit, a synergistic effect was observed in only two targets, IGFBP-1 and NT-3. This highly selective upregulation of only two growth factors among 41 candidates suggests a specific molecular mechanism underlying the observed synergistic effects.

The combined extracts increased the viability of hDPC by increasing the production of IGFBP-1 and NT-3, two growth factors that are likely involved in hair growth. Based on the results of the previous study, the fact that balding dermal papilla cells have a lot less IGF-1 secretion and the proteins that bind to it shows how important the IGF-1 signaling axis is for hair follicle health and keeping the anagen phase going [34]. New research on human umbilical cord blood mesenchymal stem cells (hUCB-MSCs) showed that these cells greatly increase hair growth by improving the viability of dermal papilla cells and the main growth factor, IGFBP-1, related to hair growth. These results support our observation that IGFBP-1 plays a critical role in modulating IGF-1 bioavailability, suggesting its potential as a therapeutic target for hair follicle regeneration [35]. Our findings align with the growing evidence that neurotrophins play a crucial role in hair follicle biology. Our study showed that increasing NT-3 may help hair growth by encouraging keratinocyte proliferation and controlling apoptosis. This agrees with earlier research that said NT-3 and its receptor TrkC were important for the growth and remodeling of hair follicles [36]. NT-3 is expressed at different times during the hair cycle and is important for controlling follicular regression. This suggests that it promotes growth and controls hair cycling transitions [37]. Additionally, studies have demonstrated that TrkA agonists such as gambogic amide safeguard the pigmentation of hair follicles and stimulate their growth in vitro. This shows that targeting neurotrophin signaling pathways could be a more general way to treat hair growth and pigmentation problems [38]. The simultaneous upregulation of both IGFBP-1 and NT-3 suggests a potential coordinated mechanism in promoting hair growth, where IGFBP-1 may enhance IGF-1 signaling while NT-3 supports keratinocyte function, potentially creating a synergistic micro-environment for hair follicle development.

The ingredient analysis identified genistein from RN and 2354-T2G from PM as the active compounds responsible for these effects, corroborating prior research that associates these compounds with hair growth and inhibition of hair loss [18,39–41]. When compounds were each treated at a 10~20 μM concentration range, increases in hDPC proliferation were observed (Fig 6). However, in the combined extract (RN at 20 μg/mL and PM at 5 μg/mL), each compound's level is at least six to over one hundred-fold lower than the single-compound concentrations (Table 1). Despite these reduced levels, the extract still significantly promoted hDPC proliferation, likely through synergistic mechanisms at lower

concentrations. These findings suggest that multiple plant-derived molecules can achieve similar or greater efficacy than single-compound approaches, yet at much lower concentrations. In the combined extract, various compounds appear to interact through distinct pathways to enhance their collective effect. Rather than seeking direct equivalence, the critical observation is that a notable effect emerges even at substantially reduced active concentrations, supporting the possibility of synergy. Future research should further elucidate these interactions and optimize compound ratios for enhanced efficacy.

## Conclusions

This research indicated that the combined extracts of RN and PM at a 4:1 ratio considerably improve hDPC proliferation and viability, even under testosterone-induced cytotoxicity, positioning them as prospective agents for promoting hair development. The observed synergistic effects were partially ascribed to elevated release of IGFBP-1 and NT-3, essential growth factors associated with hair follicle development. Active constituents, including genistein from RN and 2354-T2G from PM, were recognized as factors contributing to this impact. These findings indicate that RN and PM extracts may serve as natural therapeutic agents for hair loss; nevertheless, more study, especially clinical validation, is required to establish their efficacy and elucidate their molecular pathways *in vivo*.

## Supporting Information

**S1 File.** S1 Data. (1–3) Raw growth factor array, western blot image. S2 Data. Raw data in the manuscript. (ZIP)

## Acknowledgments

We appreciate Enago, an editing brand of Crimson Interactive Inc., for English proofreading. The authors would also like to thank The National Instrumentation Center for Environmental Management (NICEM) at SNU for their assistance in HPLC analysis and BOBSNU Co. Ltd. for manufacturing and providing sample extracts.

## Author contributions

**Conceptualization:** Jiwon Seo, Chang Hyung Lee.

**Data curation:** Jiwon Seo, Chanhyeok Jeong, Chang Hyung Lee.

**Formal analysis:** Jiwon Seo, Chanhyeok Jeong, Chang Hyung Lee.

**Funding acquisition:** Chang Hyung Lee, Ki Won Lee.

**Methodology:** Jiwon Seo, Chanhyeok Jeong, Chang Hyung Lee.

**Writing – original draft:** Jiwon Seo, Chang Hyung Lee.

**Writing – review & editing:** Jung Han Yoon Park, Chang Hyung Lee, Ki Won Lee.

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
