## [Decision Letter · Decision Letter 0]

2 Jan 2025

PONE-D-24-45676Synergistic effects of Rhynchosia nulubilis and Polygonum multiflorum extract combination on cell proliferation via targeting IGFBP-1 & NT-3 and cytotoxicity suppression in testosterone-induced human dermal papilla cellsPLOS ONE

Dear Dr. Lee,

Thank you for submitting your manuscript to PLOS ONE. After careful consideration, we feel that it has merit but does not fully meet PLOS ONE’s publication criteria as it currently stands. Therefore, we invite you to submit a revised version of the manuscript that addresses the points raised during the review process.

We look forward to receiving your revised manuscript.

Kind regards,

Sekyu Choi

Academic Editor

PLOS ONE

Journal Requirements:

This work was supported by BOBSNU Co. Ltd., Bio & Medical Technology Development Program of the National Research Foundation (NRF), funded by the Korean government (MSIT) (No. 2020M3H1A1073304), and the National Research Foundation of Korea(NRF) grant funded by the Korea government(MSIT) (RS-2024-00333238).  

5. We note that your Data Availability Statement is currently as follows: All relevant data are within the manuscript and its Supporting Information files.

Reviewers' comments:

Reviewer's Responses to Questions

**Comments to the Author**

1. Is the manuscript technically sound, and do the data support the conclusions?

Reviewer #1: Yes

Reviewer #2: No

2. Has the statistical analysis been performed appropriately and rigorously? 

Reviewer #1: Yes

Reviewer #2: Yes

3. Have the authors made all data underlying the findings in their manuscript fully available?

Reviewer #1: Yes

Reviewer #2: Yes

4. Is the manuscript presented in an intelligible fashion and written in standard English?

Reviewer #1: Yes

Reviewer #2: Yes

5. Review Comments to the Author

Reviewer #1: This is a well-executed study that makes a meaningful contribution to our understanding of Synergistic effects of Rhynchosia nulubilis and Polygonum multiflorum extract combination on cell proliferation via targeting IGFBP-1 & NT-3 and cytotoxicity suppression in testosterone-induced human dermal papilla cells. While I recommend this paper for publication, I suggest addressing several minor points to further strengthen the manuscript.

- Author should explain about the background of combined extract ratio. PM obviously present the beneficial effect more than RN, Why were they combined?

- In case of testosterone-induced cytotoxicity study, positive control should be finasteride regarding the mechanism of action.

- In the study of each active compounds affecting hDPC proliferation under testosterone conditions, is each concentration of active compounds correlated or equivalent to the concentration of each active compound in combined extract? please identify.

- All picture should increase the quality.

Reviewer #2: This study showed that the combined extracts of Rhynchosia nulubilis (RN) and Polygonum multiflorum (PM) at a 4:1 ratio considerably improve hDPC proliferation and viability, even under testosterone-induced cytotoxic environments. The observed synergistic effects were partially ascribed to elevated release of IGFBP-1 and NT-3, essential growth factors associated with hair follicle development. Active constituents, including genistein from RN and 2354-T2G from PM, were recognized as factors contributing to this impact. However, we still have many concerns and is not suitable for publication in this journal:

1, The extracts of Rhynchosia nulubilis (RN) and Polygonum multiflorum (PM) has complicated ingredients, and the result of treating the complex ingredient decoction in the cell experiment is not rigorous.

2, Why was 4:1 chosen instead of some other ratio, and the paper does not provide experimental evidence.

3, In Figure 4A, there are several peaks with high response at 6-8min retention time. What are they? Why weren't they identified in subsequent studies?

4, In Figure 4, The compounds b,c,e and f were not isolated at baseline, the content of these compounds are not accurate in Table 1

6. PLOS authors have the option to publish the peer review history of their article (what does this mean? ). If published, this will include your full peer review and any attached files.

**Do you want your identity to be public for this peer review?** For information about this choice, including consent withdrawal, please see our Privacy Policy .

Reviewer #1: **Yes: ** Wudtichai Wisuitiprot

Reviewer #2: No

---

## [Author Response · Author response to Decision Letter 1]

13 Feb 2025

Academic Editor Comments to Author:

Comment 1: When submitting your revision, we need you to address these additional requirements. Please ensure that your manuscript meets PLOS ONE's style requirements, including those for file naming.

Response 1: Thank you very much for your comments. We carefully revise my manuscript to meet PLOS ONE’s style requirements, including file naming, using the templates provided at the links you shared.

Comment 2: PLOS ONE now requires that authors provide the original uncropped and unadjusted images underlying all blot or gel results reported in a submission’s figures or Supporting Information files.

Response 2: Thank you for the information. We provided images as required and ensured our figures meet all PLOS ONE guidelines. We also noted in our cover letter where these images are located.

Comment 3: PLOS requires an ORCID iD for the corresponding author in Editorial Manager on papers submitted after December 6th, 2016. Please ensure that you have an ORCID iD and that it is validated in Editorial Manager.

Response 3: Thank you for comments. I validated my ORCID iD in Editorial Manager as instructed.

Comment 4: Please state what role the funders took in the study.

Response 4: Thank you for informing us. The funders were not involved in the study design, data collection and analysis, publication decision, or manuscript preparation. We have also added this statement to the Funding section as requested.

Comment 5: We note that your Data Availability Statement is currently as follows: All relevant data are within the manuscript and its Supporting Information files.

Please confirm at this time whether or not your submission contains all raw data required to replicate the results of your study. Authors must share the “minimal data set” for their submission. PLOS defines the minimal data set to consist of the data required to replicate all study findings reported in the article, as well as related metadata and methods. If your submission does not contain these data, please either upload them as Supporting Information files or deposit them to a stable, public repository and provide us with the relevant URLs, DOIs, or accession numbers. For a list of recommended repositories, please see https://journals.plos.org/plosone/s/recommended-repositories.

Response 5: We confirm that our submission contains the complete minimal data set required to replicate the study's results. All underlying raw data including values behind means and standard deviations, data points used to build graphs, and points extracted from images are provided in the Supporting Information files and deposited in a stable public repository.

Reviewer(s)' Comments to Author:

Reviewer 1 Comments:

Comment 1: Author should explain about the background of combined extract ratio. PM obviously present the beneficial effect more than RN, why were they combined?

Response 1: We appreciate the reviewer's question regarding the rationale behind our combination strategy. To better demonstrate our experimental approach and findings, we have added new data (Fig 1b) and reorganized our figures to show a more logical progression of our study. First, we investigated the individual effects of RN and PM extracts on hDPC proliferation under both normal and testosterone-induced conditions (Fig 1). While both extracts showed concentration-dependent effects under normal conditions, with PM demonstrating stronger effects at higher concentrations, their protective effects under testosterone-induced conditions were more limited. Specifically, RN showed no significant protective effects at all concentrations (2.5-80 μg/mL), while PM demonstrated effectiveness across a broader range (10-80 μg/mL) (Fig 1b). Based on these observations, we investigated whether combining these extracts at lower, minimally effective concentrations could produce synergistic effects. Through systematic testing of various combinations under testosterone-induced conditions, we found that a 4:1 ratio (RN 20 μg/mL : PM 5 μg/mL) demonstrated exceptional protective effects, significantly increasing cell viability from 52% to 78% (Fig 2). This synergistic effect was further validated by the enhanced secretion of growth factors IGFBP-1 and NT-3 (Fig 3, 4), which were not significantly elevated by individual treatments.This approach was intentionally designed to maximize therapeutic effects while minimizing the required amount of each extract, particularly the more potent PM extract, leading to a more efficient formulation. In addition, we included a more detailed explanation of synergy and our methodological approach in two additional paragraphs within the revised Discussion section. These paragraphs outline how we identified synergy, the specific approach to demonstrate it, and the implications for optimizing the combined extracts for maximal therapeutic effect (revised manuscript with track changes file, line 450-470).

Comment 2: In case of testosterone-induced cytotoxicity study, positive control should be finasteride regarding the mechanism of action.

Response 2: We appreciate the reviewer's comment regarding finasteride. While finasteride's mechanism through 5α-reductase inhibition is relevant to testosterone-mediated hair loss, our choice of minoxidil as a positive control was deliberately aligned with our research goal of developing topical applications for hair loss treatment. Minoxidil, as a widely used topical treatment, was selected considering our focus on developing cosmeceutical products such as hair care formulations. Additionally, minoxidil's reported effects on vasodilation and protein kinase activation suggested potential enhancement of growth factor secretion, although our results showed unexpected outcomes under testosterone-induced conditions. This finding provides valuable insights into the complex interactions between different hair loss treatment mechanisms and hormonal environments. While we acknowledge that finasteride could provide additional mechanistic insights, our selection of minoxidil as a positive control reflects our specific aim of developing topical treatments for hair loss. We revised one paragraph in the Discussion section to better clarify the rationale for using positive control in this model and to enhance readers’ understanding of its role (revised manuscript with track changes file, line 422-434).

Comment 3: In the study of each active compounds affecting hDPC proliferation under testosterone conditions, is each concentration of active compounds correlated or equivalent to the concentration of each active compound in combined extract? please identify.

Response 3: We appreciate the reviewer's question about our approach to active compounds. Based on previous literature, we first identified daidzein, glycitein, and genistein as known bioactive compounds from RN [Lee et al., 2017], and 2354-T2G, E8G, and P8G were identified as major components of PM according to the Korea Ministry of Food and Drug Safety. Several of these compounds have shown beneficial effects on various cell types including hDPCs in previous studies [Li et al., 2015; Zhao et al., 2011; Qian et al., 2020; Bensaada et al., 2022]. In our study, we confirmed the presence and quantities of these compounds in our extracts through HPLC analysis and extended previous findings by testing these compounds specifically in a testosterone-induced cytotoxicity model. Our results revealed that genistein from RN and 2354-T2G from PM showed protective effects under testosterone-induced stress. The concentration ranges (5-20 μM) were selected based on previously reported effective doses in various bioactivity studies [Yang et al., 2019; Sun et al., 2013]. Our findings newly demonstrate that these known bioactive compounds are also effective in protecting hDPCs against testosterone-induced damage, providing additional insights into their potential mechanisms of action in hair loss prevention. Additionally, we revised the Discussion section to further expand on the implications suggested by the active compound's results and to address their potential limitations (revised manuscript with track changes file, line 501-514).

Lee, C.C., et al., Comprehensive phenolic composition analysis and evaluation of Yak-Kong soybean (Glycine max) for the prevention of atherosclerosis. Food chemistry, 2017. 234: p. 486-493.

Li, Y., et al., Hair growth promotion activity and its mechanism of Polygonum multiflorum. Evidence‐Based Complementary and Alternative Medicine, 2015. 2015(1): p. 517901.

Zhao, J., et al., Dietary isoflavone increases insulin-like growth factor-I production, thereby promoting hair growth in mice. The Journal of Nutritional Biochemistry, 2011. 22(3): p. 227-233.

Qian, J., et al., A review on the extraction, purification, detection, and pharmacological effects of 2, 3, 5, 4’-tetrahydroxystilbene-2-O-β-d-glucoside from Polygonum multiflorum. Biomedicine & Pharmacotherapy, 2020. 124: p. 109923.

Bensaada, S., et al., Development of an Assay for Soy Isoflavones in Women’s Hair. Nutrients, 2022. 14(17): p. 3619.

Yang, H., et al., Orobol, an enzyme-convertible product of genistein, exerts anti-obesity effects by targeting casein kinase 1 epsilon. Scientific Reports, 2019. 9(1): p. 8942.

Sun, Y.N., et al., Promotion effect of constituents from the root of Polygonum multiflorum on hair growth. Bioorganic & medicinal chemistry letters, 2013. 23(17): p. 4801-4805.

Comment 4: All picture should increase the quality.

Response 4: We appreciate the reviewer's comment regarding figure quality. All figures have been regenerated at 300 dpi resolution in TIFF format, improving clarity while maintaining the integrity of the original data. This enhancement ensures optimal visualization and consistent quality throughout the manuscript.

Reviewer 2 Comments:

Comment 1: The extracts of Rhynchosia nulubilis (RN) and Polygonum multiflorum (PM) has complicated ingredients, and the result of treating the complex ingredient decoction in the cell experiment is not rigorous.

Response 1: We appreciate the reviewer's insightful comment about the scientific rigor of evaluating synergistic effects using complex extract combinations. We acknowledge that demonstrating synergy at the molecular level would require testing specific combinations of active compounds rather than extracts.

Our study followed a systematic approach to investigate the synergistic effects. We initially established baseline effects by testing each extract individually on hDPC proliferation (Fig 1), which demonstrated clear dose-dependent responses for both RN and PM. Through subsequent testing of various extract combinations, we identified that the 4:1 (RN:PM) ratio optimally enhanced hDPC proliferation (Fig 2).

The synergistic effect was validated through multiple experimental endpoints. Beyond the enhanced cell proliferation observed compared to individual extracts, we found increased expression of specific growth factors (IGFBP-1 and NT-3) that were not significantly elevated by individual extracts alone (Fig 3,4). To understand the molecular basis of these effects, we identified and quantified key compounds (genistein from RN and 2354-T2G from PM) (Fig 5) and demonstrated their individual efficacy under testosterone-induced conditions (Fig 6). We acknowledge that future studies should investigate specific molecular interactions between identified active compounds at various ratios, establish structure-activity relationships for the observed synergy, and examine potential signaling pathway interactions. While our current approach using extract combinations may have limitations in terms of molecular precision, it provides strong evidence for enhanced efficacy through combined treatment. This approach also aligns with traditional herbal medicine practices where complex mixtures often show beneficial effects through multiple compound interactions. We revised the Abstract, Introduction, and Discussion sections to reflect these considerations (revised manuscript with track changes file, line 48-65, 86-97, 108-118, 152-155, 159-173, 450-514). We would greatly appreciate it if you could kindly review the updated sections.

Comment 2: Why was 4:1 chosen instead of some other ratio, and the paper does not provide experimental evidence.

Response 2: We appreciate the reviewer's question regarding the selection of our 4:1 ratio. To better address this concern and provide experimental evidence, we have added new data showing the concentration-dependent effects of individual extracts under testosterone-induced conditions (Fig 1b). Our results demonstrate that RN did not show significant protective effects at all concentrations, while PM demonstrated effectiveness across a broader range (10-80 μg/mL). We examined the combination of two distinct extracts, RN and PM, by considering two main strategies: increasing the ratio of RN or increasing the ratio of PM. In the experiment, the ratio of extract RN, which showed less efficacy at lower concentrations compared to extract PM, was slowly raised from 1:1 to 8:1 to test the synergistic effect. Synergy was initially observed at a 2:1 ratio, with a markedly enhanced effect at a 4:1 ratio. The efficacy at a ratio of 8:1 was slightly lower than at 4:1, suggesting a possible saturation effect of synergism (Figure 2). Subsequently, a growth factor array analysis was performed to identify the specific targets that were upregulated under the 4:1 condition. We have added a new paragraph in the Discussion section to reflect these points, and we would greatly appreciate your review (revised manuscript with track changes file, line 450-479).

Comment 3: In Figure 4A, there are several peaks with high response at 6-8min retention time. What are they? Why weren't they identified in subsequent studies?

Response 3: Thank you for your insightful comment. The peaks observed at 6–8 minutes retention time in Figure 5A (previously labeled as Figure 4A) were identified as compounds Daidzin (RT: 6.77 min) and Genistin (RT: 7.30 min). These compounds had already been characterized in previous studies on the extract. However, they were not included in subsequent analyses as they were not among the active components reported to have significant hair loss treatment effects. We prioritized compounds with well-documented efficacy for hair loss improvement based on previous literature. In follow-up experiments, these particular compounds were found to have minimal or no effects on hair loss improvement, which led to their exclusion from the list of major active components. We appreciate the opportunity to clarify this point and emphasize it in the revised manuscript (revised manuscript with track changes file, line 501-514).

Comment 4: In Figure 4, The compounds b,c,e and f were not isolated at baseline, the content of these compounds are not accurate in Table 1

Response 4: Thank you for raising this important point regarding the separation of compounds in Figure 5 (previously labeled as Figure 4A). Compounds b, c, e, and f were indeed separated from the baseline. Although their concentrations were relatively low, we obtained standard compounds to verify their retention times (RT) and further confirmed their identities as single peaks using mass spectrometry. These additional steps ensured the reliability of the analysis and validated the quantitative data presented in Table 1. Specifically, the compounds identified in the RN extract were as follows: a (Daidzein, M.W: 254.23, RT: 8.16 min), b (Glycitein, M.W: 284.26, RT: 8.37 min), and c (Genistein, M.W: 270.24, RT: 9.18 min). Similarly, the compounds identified in the PM extract were: d (2354-T2G, M.W: 406.39, RT: 7.14 min), e (E8G, M.W: 432.38, RT: 8.02 min), and f (P8G, M.W: 446.41, RT: 8.44 min).

---

## [Decision Letter · Decision Letter 1]

12 Mar 2025

Synergistic effects of Rhynchosia nulubilis and Polygonum multiflorum extract combination on cell proliferation via targeting IGFBP-1 & NT-3 and cytotoxicity suppression in testosterone-induced human dermal papilla cells

PONE-D-24-45676R1

Dear Dr. Lee,

We’re pleased to inform you that your manuscript has been judged scientifically suitable for publication and will be formally accepted for publication once it meets all outstanding technical requirements.

Kind regards,

Sekyu Choi

Academic Editor

PLOS ONE

Additional Editor Comments (optional):

Reviewers' comments:

Reviewer's Responses to Questions

**Comments to the Author**

1. If the authors have adequately addressed your comments raised in a previous round of review and you feel that this manuscript is now acceptable for publication, you may indicate that here to bypass the “Comments to the Author” section, enter your conflict of interest statement in the “Confidential to Editor” section, and submit your "Accept" recommendation.

Reviewer #1: All comments have been addressed

Reviewer #2: All comments have been addressed

2. Is the manuscript technically sound, and do the data support the conclusions?

Reviewer #1: Yes

Reviewer #2: Yes

3. Has the statistical analysis been performed appropriately and rigorously? 

Reviewer #1: Yes

Reviewer #2: Yes

4. Have the authors made all data underlying the findings in their manuscript fully available?

Reviewer #1: Yes

Reviewer #2: Yes

5. Is the manuscript presented in an intelligible fashion and written in standard English?

Reviewer #1: Yes

Reviewer #2: Yes

6. Review Comments to the Author

Reviewer #1: (No Response)

Reviewer #2: Since the authors have made the necessary changes to the questions we have raised, we recommend acceptance of the paper.

7. PLOS authors have the option to publish the peer review history of their article (what does this mean? ). If published, this will include your full peer review and any attached files.

**Do you want your identity to be public for this peer review?** For information about this choice, including consent withdrawal, please see our Privacy Policy .

Reviewer #1: No

Reviewer #2: No

---

## [Editor Report · Acceptance letter]

PONE-D-24-45676R1

PLOS ONE

Dear Dr. Lee,

I'm pleased to inform you that your manuscript has been deemed suitable for publication in PLOS ONE. Congratulations! Your manuscript is now being handed over to our production team.

Kind regards,

on behalf of

Dr. Sekyu Choi

Academic Editor

PLOS ONE